# Digitalization and Strategic Changes in Romanian Retail Fuel Networks: A Qualitative Study

**Dan Andrei Panduru * and Cezar Scarlat**

Doctoral School of Entrepreneurship, Business Engineering & Management, University Politehnica of Bucharest, 060042 Bucharest, Romania
* Correspondence: dpanduru91@gmail.com; Tel.: +40-727-802-624

**Abstract:** The oil and gas industry is among the most affected industries as a result of war in Ukraine, on top of other economic, political, and environmental global turbulences that culminated with the coronavirus pandemic. The purpose of this qualitative, explorative study was to identify strategic changes as well as the role played by newer technologies—digital technologies in particular—in this industry. The focus is on the Romanian oil and gas industry, more specifically on the retail fuel networks of the top companies. In addition to secondary research (literature and company documents), interview-based primary research was conducted. The data were collected during spring of 2022 by conducting interviews with two groups of subjects: the strategists—consisting of top managers from the largest companies active in the oil and gas industry in Romania; and the informed consumers—selected from people working in the oil and gas industry. The interview guides were slightly different depending on the two groups targeted, and the structure of the interview guide was developed according to research questions. Among the findings, we can observe that the fuel retail market and consumer behaviour changed due to a series of factors, such as the global economic crisis, COVID-19, the Russian invasion of Ukraine, and inflation. Those factors forced fuel retail companies, at the global level, to invest in filling station shops, services development, digitalization, and divestment—selling gas station networks in countries with poor integration with refineries. Romanian fuel retail companies are following the global trends and focusing on filling station shops, alternative fuels development, and digitalization. The results are followed by discussions, and several managerial implications are suggested. The study limitations and several further research paths are also identified. Based on the data available, we can conclude that the strategic directions at the level of products and services are aligned, but at the execution level, specialists offer different solutions for customer expectations.

**Keywords:** strategic changes; digitalization; digital strategy; digital transformation; digital acceleration; coronavirus pandemic as digital accelerator; fuel retail networks; trends in oil and gas industry

## 1. Introduction: Global Changes

### 1.1. Global Strategic Shift

The concept of *globalisation* is about a half-century old as it was launched [1] by French economist François Perroux in the 1960s (as *mondialisation*), and it was solidly coined and spread by Theodore Levitt during the 1980s. The concept evolved from globalisation to *glocalisation* [2] to current "globalisation reinvented" as *The Economist* [3] defines the current trend. The need for reinventing globalisation is a result of a series of global turbulences (socioeconomic, political, and environmental) that recently culminated with the coronavirus pandemic and war in Ukraine [4] and its impact on the oil and gas industry [5].

Traditional globalisation looked for efficiency; however, "Hyper-efficient globalization also had problems. Volatile capital destabilised financial markets. Many blue-collar workers in rich countries lost out." [3] (p. 9). Currently, the same source identifies two more serious

issues: (i) some lean supply chains "are not as good value as they appear"; and (ii) increased dependency on "autocracies that abuse human rights and use trade as a means of coercion".

As a result, the *companies shift from efficiency to security and resiliency* [6]. The best indicator is that inventories of the largest 3000 companies worldwide increased significantly since 2016: from 6% up to 9% [3] (p. 9)—as precautionary measures.

*1.2. Impact on Strategic Thinking*

The aforementioned turbulences impacted the companies' strategic thinking; the evolution of strategic thinking experienced a crossroads—from which two tendencies could be identified [7,8]—apparently contradictory:

- Shorter term, more flexible strategies—based on new concepts as *blue ocean* [9] or *marketing as strategy* [10];
- Very long-term *foresight* exercises [11,12].

The two trends do not contradict each other (but complement each other), nor are they in contradiction with previous strategies: a solid example is the use of Porter's generic strategies and/or models in the case of digital transformation projects [13]—even in the case of small businesses [14]. Therefore, the transition is neither abrupt nor total—as elements of older and newer strategies may coexist.

*1.3. Strategic Thinking and Digitalization*

Currently, the strategic thinking and digitalization are inexorably and intrinsically merging—in such a manner that discussion is about *digital strategy* [15]—small businesses included [14]. In general, the digital strategy involves *digital transformation* [16].

Authors agree that *digital transformation is driven by strategy* and not by the digital technology itself (technology is important; still, it is just a means)—according to a solid report jointly developed by *MIT Sloan Management Review* and *Deloitte* [17].

Of particular interest, is the observation that global threats—such as the coronavirus pandemic—might act, selectively, in certain circumstances, as *digital accelerators* [18,19].

When the process of digital transformation is analysed, it seems that the small and medium-size business sector (SMEs) enjoys more attention from researchers—probably for its dynamism. In addition to the examples already cited for the case of low digital maturity—when a *roadmap for the digital transformation* can be defined [14], other authors investigate the process of SME digital transformation from different perspectives—e.g., from the capability perspective [20] or from the ecosystemic perspective [21]. Of note is that digital marketing is currently a common approach [22].

The main research question is *if* and *how* the oil and gas industry has reacted and how sensitive it was to these strategic changes. As there seems to be a literature gap in this respect, this study aims to contribute to covering this gap: how the Romanian oil and gas industry was impacted—in particular, retail fuel networks belonging to top companies active in this industry. In addition, the process of digitalization (digital transformation) is under scrutiny as well.

Consequently, this paper is structured as follows: a closer look at the oil and gas industry—in global and European/Romanian contexts—supported by secondary research; methodology—method, instrument, and data (primary research on retail fuel networks of the top oil and gas companies); results and discussions (research limitations and future research avenues); followed by conclusions. Managerial implications are of interest for the main stakeholders (industry strategists and companies' top managers).

## 2. Strategic Changes in Oil and Gas Industry
*2.1. General Trends in Oil Industry*

In the past years, many factors influenced the fuel retail market globally. Up to today, the global economic crisis, energy transition, COVID-19, the Russian invasion of Ukraine, followed by inflation dramatically affected the market and consumer behaviour.

Taking into consideration these factors, fuel retail companies find themselves in a sensitive position and have to adapt and invest to retain and attract customers to survive.

Consumption of petroleum products in Europe had an upward trend during the years 1990–2006. Starting with 2006, due to the global economic crisis, it started to decrease until 2014 (Figure 1).

The increase in consumption starting in 2014 was significantly negatively affected in 2019 by the COVID-19 pandemic [23].

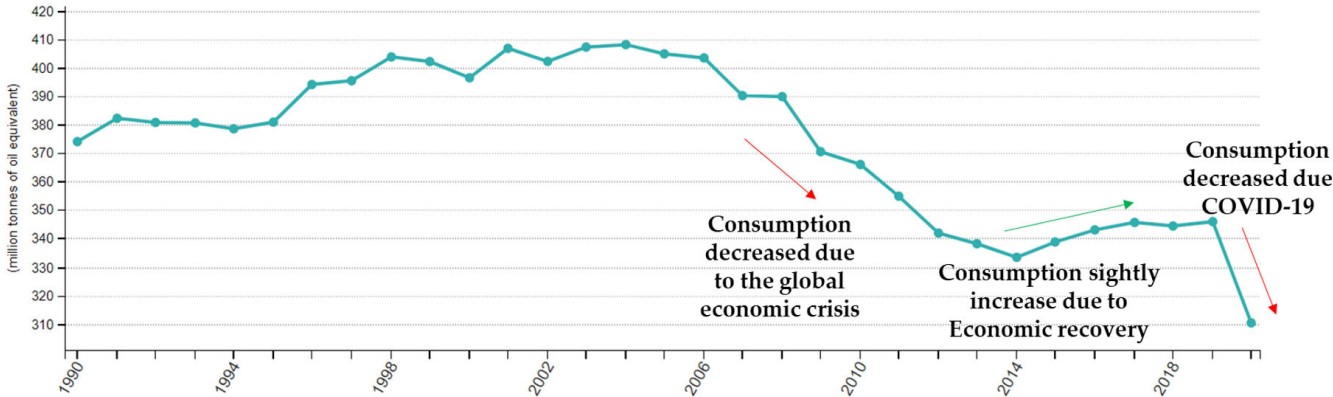

**Figure 1.** Final consumption (energy use), oil and petroleum products, in the EU, 1990–2020. Source: Eurostat 2022 [24].

In Europe, the gross profit of fuel retail companies comes from both the sale of fuel (between 60 and 70%) and non-fuel products (between 40 and 30%) [24]. Considering the significant percentage of gross profit from the area of non-fuel products, the trend in fuel chains is to focus on quality, fresh, and healthy food choices using information gathered about consumer behaviour [25].

Both the economic crisis and COVID-19 reduced the demand for oil and gas products, which generated a price drop in the pump price. By end of 2021, the demand for fuel products exceeded supply, followed by the Russian sanctions that generated a shortage in supply, and the price per barrel, followed by the fuel pump price, dramatically increased (Figure 2).

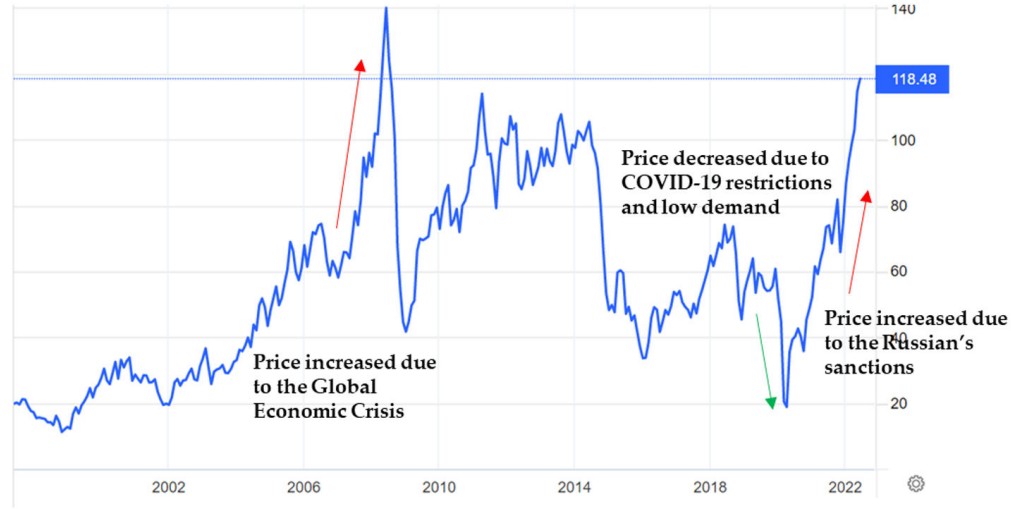

**Figure 2.** Oil prices over the past two decades (2002–2022). Source: tradingeconomics.com [26].

### 2.2. Changes in Oil and Gas Industry—In Particular in Fuel Distribution

The main strategic trends of the fuel retail companies at the global level are focusing on alternative fuels, digitalization, convenience stores or multi-brand partnership, and connected services [27].

In terms of digital transformation, one of the main goals of oil and gas companies is to develop contactless payment systems. By developing contactless payment features, fuel networks aim to streamline point-of-sale financial operations and increase customer satisfaction by reducing time spent at cash registers and improving their experience [28].

A McKinsey research report demonstrates that *digital developments are an important factor in the strategies of fuel distribution companies* and have a significant positive impact on market differentiation. Digital developments have the potential to increase customer satisfaction by up to 20%, reduce operating costs by 20 to 40%, and improve conversion rates by up to 20% [29].

Other researchers argue that the oil and gas industry is constantly looking for innovative solutions to streamline business processes, reduce costs, and improve safety at work [30]. The focus of oil and gas retail networks is on predicting demand and forecast trends in retail using solutions based on artificial intelligence and machine learning technologies, big data analytics to design the most personalised offers for customers, and also, the Internet of things (IoT) for security.

At the level of the oil and gas industry, artificial intelligence is used in extraction technology, in the production and marketing of petroleum products, and to increase the accuracy of the data collected [31].

Artificial intelligence technology is also used for smart payments [28]—which is one of the strategic directions on which fuel retail networks focus to streamline business operations and maximise the time that the consumer can use on-station [32]. The type of smart payments that companies in the industry focus on is integrated license plate recognition systems, smart cash registers, or fuel pump payments. Moreover, companies focus on smart loyalty programs, which are tools that fuel chains use to reward loyal consumers. The strategic trends of companies in the industry, but also one of the main expectations of consumers, refer to these loyalty programs. Personalised loyalty programs can be developed using big data analysis by collecting consumer data from cards used for refuelling and associated with cash register information on purchases of non-fuel products and services. Through big data, companies can capture consumer behaviour and generate personalised loyalty offers.

Gupta and Gergele have shown that to attract future consumers, fuel retail networks should focus on [24]:

(i) Creating a personalised sales experience in each location by integrating available consumer data (financial data, social media data to demographics, and consumer revenue data) to build personalised loyalty offerings and sales experiences;

(ii) Accessing alternative fuels through the development of electric charging points, hydrogen supply points, but also other derivatives of petroleum products, such as liquefied petroleum gas, compressed natural gas, and liquefied natural gas.

Personalised offerings and assortment in gas station stores, built on digitally collected data, and complemented by a personalised loyalty program, will bring considerable benefits to companies holding fuel networks [McKinsey, 2020]. Also of great interest, are multi-brand partnerships, partnerships in which fuel retail networks share their locations with companies in the retail industry, such as *Carrefour*, *Auchan*, *Starbucks*, and *Subway*, to attract new consumers.

From the consumers' perspective, post-COVID-19 trends show that they are interested in issues such as [27]:

- Sustainability and the environment, the comfort to be supported by companies through fast operations;
- Hybrid integration of the physical environment with the virtual environment, through which companies allow consumers to access physical experiences through virtual digital applications;
- Flexibility and minimisation of the time spent by the consumer on the operations of the companies to facilitate the maximisation of his free time;
- Representation of consumer interests by companies through media and social networks, thus generating the feeling of belonging to a cause, situation, and brand.

### 2.3. Strategic Changes in Romanian Retail Fuel Networks

Romania is following the global trends in terms of fuel retail trends. The main players in the Romanian fuel retail market are *OMV Petrom*, *Rompetrol*, *Mol*, and *Lukoil*.

*OMV Petrom* has the largest market share, amounting to approximately 40%, followed by *Rompetrol* with a total market share of 24%, *Lukoil* at 22%, and *Mol* with 15% (Figure 3).

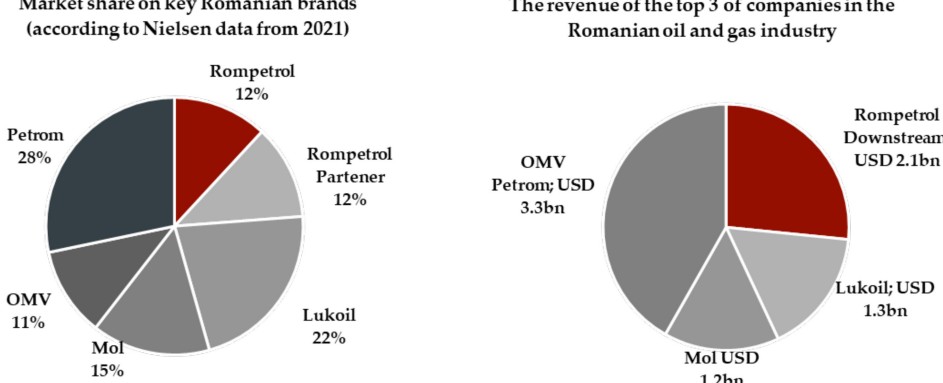

**Figure 3.** The main fuel retail companies in Romania and their market share. Source: based on market data from Nielsen, 2021 [33].

In terms of strategic trends, the *OMV Group* aims to eliminate carbon emissions by 2050 and focuses on the development of alternative fuels such as biofuels. It is also developing new strategic directions for geothermal energy and carbon capture and storage [34].

The *Mol* company's strategy is two-pronged: the transition from oil to chemicals and an integrated digitization-focused mobility service provider. *Mol* focuses on the development of the service area (the newest service is individual laundry), the variety of non-fuel products, but also the portfolio of alternative fuels [35].

*Rompetrol*'s strategy focuses on expansion, intending to open 44 new fuel stations by 2024. Another strategic project is the partnership with the *Carrefour* group, which operates several stores in the *Rompetrol* station network. Other strategic directions of *Rompetrol* are represented by alternative fuels, the development of the electric charging network, and the development of digital solutions for the non-fuel area (pre-order applications, applications for loyalty programs, etc.) [36].

The Romanian retail market, driven by increased consumption and economic growth, had one of the best dynamics in the region. *Petrol Stations* experienced an increase in the revenue stream from complementary products (food/drinks/sandwiches/coffee, etc.). Even though sales generated by *Petrol Stations* are only around 2% of the total FMCG (fast moving consumer goods) market, the growth of this distribution channel (from 1% in 2019) shows important potential and customer orientation towards convenience (Figure 4).

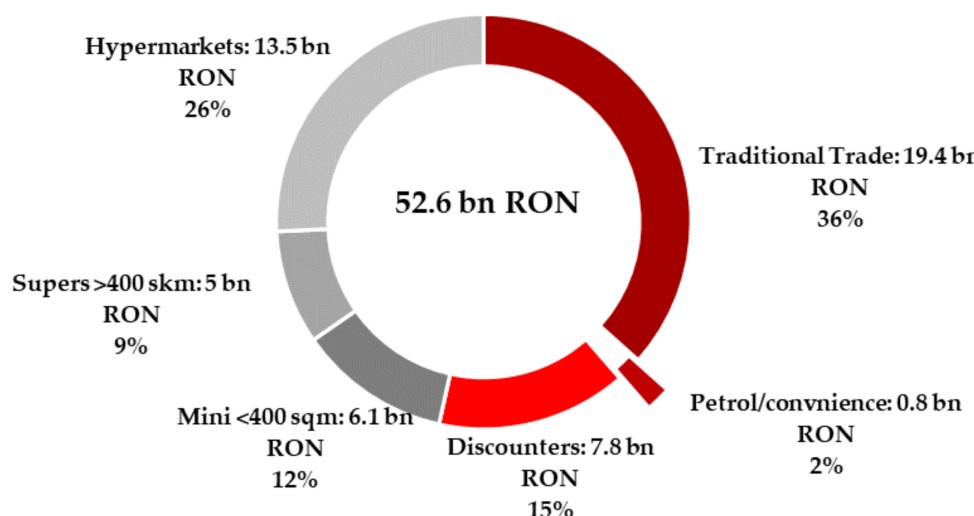

**Figure 4.** Value of the total FMCG (fast moving consumer goods) industry, in Romania, Q3 2021. Source: adapted after Nielsen [33].

### 3. Materials and Methods

This paper presents just a part of a larger research project; it is a qualitative, explorative study—based on secondary research (studies and company documents) as well as primary research: qualitative methodology, using a semi-structured interview with oil and gas industry specialists and consumers of oil and non-fuel products. Semi-structured interviews are flexible and facilitate the opportunity to capture "hidden facets of human and organizational behavior" [37].

The data were collected during spring of 2022 (March–May) by conducting interviews with two groups of subjects: the strategists (S)—consisting of top managers from the largest companies active in the oil and gas industry in Romania; and the informed consumers (C)—selected from people working in the oil and gas industry.

There are four basic research questions to be addressed:

RQ1: Are the global and local fuel retail markets in a transition period?
RQ2: What are the main challenges for Romanian fuel retail companies?
RQ3: What will be the strategic implications of these challenges?
RQ4: How can these challenges be addressed (from a digitalization and customer expectations point of view)?

The interview guides were slightly different depending on the two groups targeted, and the general structure of the interview guide was developed according to research questions—as it is shown in Table 1.

The group of specialists (S) consists of employees of the top 3 companies in the oil and gas industry from Romania (Omv-Petrom group, Rompetrol, and Mol), with positions in the strategic department of station network development fuel sales.

The consumer group consists of people who are from the target group of the first 3 companies of fuel sales. The consumer group is built taking into account several status indicators that the first 3 companies in the oil and gas industry target as follows: age between 30 and 35, with above average incomes (Romania's average net monthly salary in 2021 was RON 3540 [38]), who constantly consume services and products from gas stations.

**Table 1.** The interview guides—adapted for the two groups targeted: company managers or strategists (S) and informed consumers (C).

| No. Crt. | Company Managers/Strategists | Consumers |
|:---:|:---|:---|
| 1 | What are the trends in the retail strategies of gas station chains? | In which direction do you think the retail strategies of the gas station chains in Romania are heading (1–3–5 years)? |
| 2 | What products and services should gas station chains develop in the future? | What would you like to find in gas stations (products, services)—what should gas stations offer to be attractive? |
| 3 | What are the expectations of customers (what they want) from a gas station chain (in terms of products and services in stations)? | What are the reasons why you choose a certain gas station chain? |
| 4 | In terms of digitization, what kind of digital applications or services should or will gas station chains offer? | In terms of digitization, what kind of digital applications or services would you like a gas station chain to offer you? |

The semi-structured interview was built on a set of predetermined questions, which offers the researcher the opportunity to exploit other questions related to the topic [39]. In this particular case, qualitative research has been used as a critical tool that provides perspectives that can challenge or support theoretical expectations [40] and provides a different perspective on the subject of the paper. The qualitative method responds to the need for research objectives by helping the current work to discover natural conditions, focusing on a holistic vision through the perception of a certain event [41].

The total number of selected respondents is 10 (five in each target group). They were selected out of the top three Romanian fuel companies.

The group of strategists includes managers in the Romanian oil and gas industry, with positions in strategic departments, responsible for developing fuel retail networks.

Note that all members of the group (C) are industry professionals that participated in the study as consumers—as depicted in Table 2.

**Table 2.** Company managers or strategists (S) and informed consumers (C)—oil and gas industry professionals—who participated in the study.

| No. Crt. | Company Managers/Strategists | Consumers |
|:---:|:---:|:---:|
| 1 | Senior executive (S1) | Salesperson (C1) |
| 2 | Trade executive (S2) | Audit professional (C2) |
| 3 | Marketing manager (S3) | Business consultant (C3) |
| 4 | Strategy developer (S4) | Marketing researcher (C4) |
| 5 | Brand manager (S5) | Advisory manager (C5) |

## 4. Results

Research of fuel retail chain strategies—from the perspectives of industry strategists and consumers—can suggest if they are aligned and calibrated, respectively. The results of the research are shown (i) from the standpoint of specialists; and (ii) from the consumers' perspective.

According to strategists (group S):

- Strategic trends of fuel retail networks incline towards expansion, service development, and positioning as convenience stores.
- Positioning as convenience stores is achieved through partnerships with market retailers (*Auchan* and *Carrefour*) or adapting the range of non-fuel products in stations and aligning product prices with those in traditional convenience stores.
- The services that fuel retail companies should develop are alternative fuels, fast payments, and services that ensure sustainability.

- The main expectations of consumers are related to the variety of fuel and non-fuel products, the prices and related services, promotions, and the friendliness of the staff.
- In terms of digitization, fuel chains offer or should offer digital applications and services—such as pre-order non-fuel products, loyalty applications, and fast payment.

According to consumers (group C):

- Strategic trends of fuel retail networks indicate a movement towards convenience stores and the development of complementary services.
- Consumers' expectations regarding shopping stations are related to the diversification of services and products (laundries, restaurants, and children's playgrounds).
- Proximity, price, and quality are the most important factors that influence consumers in choosing a particular chain of gas stations.
- From the point of view of digitalization, consumers' expectations are oriented towards fast payment and intelligent loyalty systems.

The trends of the retail strategies of the fuel retail networks viewed from the two perspectives suggest that the vision of the specialists regarding the expansion of the networks and the positioning of the fuel retail networks as convenience stores is supported by the consumer perspective. The latter argues that it is necessary to expand fuel retail networks in direct proportion to the development of transport infrastructure and the number of vehicles: "Also, with the development of infrastructure in Romania, it is necessary to ensure a sufficient number of gas stations, especially on highways, on transit routes for tourists and freight vehicles" (S3).

As for their transition to convenience stores, this view is also supported by arguments about partnerships with retail companies (C4): "the need of appearance of more and more stations with supermarket type stores (e.g., *Petrom* and *My Auchan*); soon, several fast-food restaurants with non-stop operation on the same plot with the station, not even attached, should appear".

From a service perspective, experts believe that fast payments and services that ensure sustainability will be services that companies will focus on in the future, while consumers focus more on the experience in the gas stations, leisure areas, restaurants, and car wash, but also, alternative fuels. Regarding the products, the specialists consider that own-brand products, organic products, vegan products, but also medicines should be the categories developed in the future. This aspect is also supported by consumers who consider the development of the variety of products present in the villages.

Experts in the field believe that customer expectations relate to the variety of fuel and non-fuel products, the prices of related products and services, promotions, and the friendliness of the staff.

From a digital perspective, experts believe that digital applications and services offered by fuel retail networks should focus on pre-order, non-fuel products, loyalty applications, fast payment—through smart cash registers, payment at the feed pump, or recognising the registration number of vehicles. Consumer expectations for digital services and applications focus on fast payments, smart loyalty systems, real-time fuel pricing applications, and crypto-currency purchases.

## 5. Discussion

As a research study in progress (inception phase of a larger research project), the paper has inherent limitations; the next step aims at getting more quantitative results by opening future research paths both longitudinally and transversally in the oil and gas industry: focusing on a specific case to watch the evolution of the company's strategy over time, under the influence of new technologies; and comparing current strategies of different actors active in the oil and gas industry, in a particular country (Romania).

In addition to surveying a larger representative sample—in order to conduct a multi-level quantitative research study, an appealing path to investigate is to add a third layer of respondents—regular customers. In this case the range of customer needs will definitely be

larger; in addition, the bias introduced by the "more informed customers" would be an interesting research objective itself.

Future research should also consider the dynamics of the digitalization and strategic changes in the oil and gas industry from the perspective of mentioned factors, such as the Russian invasion of Ukraine and inflation.

First and foremost, the main implication is the base for extended research. Then, the results of this study allow formulation of recommendations for the main stakeholders: longer-term for industry strategists (at the macroeconomic level) and mid-term for top managers as well as marketing specialists (at the microeconomic, company level).

## 6. Conclusions

Consumption of petroleum products in Europe has been increasing in the past 20 years. Since 2006, due to the global economic crisis, consumption has started to decline until 2014. After a short recovery period, consumption was again negatively affected in 2019 by the COVID-19 pandemic and in 2022 by the Russian invasion of Ukraine.

The main retail strategies of global fuel retail companies focus on innovation through digital solutions, expanding supply chains, and reducing costs by selling assets in certain markets.

At the local level, the main retail strategies of Romanian fuel retail companies follow global trends and focus on expansion, investments in the redevelopment of the shops, continuous development of services and product variety, loyalty programs, and investments in non-fuel services. Strategic trends focus on migrating retail chains to convenience stores through the partnership model or developing a variety of products and calibrating prices with consumer needs.

In conclusion, by analysing the data collected from the specialised reports and comparing them with the data collected from the interviews organised with both specialists in the field and with consumers, we can notice that the strategic directions at the level of products and services are aligned (network expansion, payments, transition to convenience stores, and availability of alternative fuels), but at the execution level, specialists offer different solutions for customer expectations. In other words, the system of fast payments for optimising payment transactions and maximising the time spent on-location at the consumer's choice is taken into account both in specialist reports and by specialists and consumers. The way it is applied by specialists is different from the expectations of consumers. Consumers want a fast payment system at the pump, and experts believe that the fast payment solution refers to smart cash registers.

At the level of need, companies strive to cover what the market and global trends dictate, but at the level of functionality, companies' solutions are different from the needs of consumers themselves.

Following the research questions stated in the paper, we can observe that the oil and gas industry is in a transition period generated by several factors mentioned in the paper, such as the global economic crisis, COVID-19, the Russian invasion of Ukraine, and inflation.

Those factors challenged the industry by generating an increase in fuel pump price, a decrease in volume consumption, and a change in the consumers' behaviour that is forcing fuel retail companies, at the global and local level, to invest in filling station shops, new services development, digitalization for customer retention, and attracting new ones.

In facing the challenges of these factors, oil and gas retailers must properly address them in such a way as to manage covering the new needs of the consumers.

The strategic implication of these challenges is significant, and improper treatment can have a negative impact at the company and market level.

**Author Contributions:** Conceptualization, D.A.P. and C.S.; methodology, D.A.P.; formal analysis, D.A.P.; investigation, D.A.P. and C.S.; resources, D.A.P. and C.S.; data curation, D.A.P.; writing—original draft preparation, D.A.P.; writing—review and editing, D.A.P. and C.S.; visualization, D.A.P.; supervision, C.S.; project administration, C.S.; funding acquisition, not applicable. All authors have read and agreed to the published version of the manuscript.

**Funding:** This research received no external funding.

**Institutional Review Board Statement:** Not applicable.

**Informed Consent Statement:** Informed consent was obtained from subjects involved in the study.

**Data Availability Statement:** The data presented in this study are available on request from the corresponding author. The data are not publicly available due to GDPR policy.

**Conflicts of Interest:** The authors declare no conflict of interest.

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
