# Peer review of "Digitalization and Strategic Changes in Romanian Retail Fuel Networks: A Qualitative Study"

_information, doi:10.3390/info13090416_

Round 1

Reviewer 1 Report

The study is devoted to a relevant topic and may potentially be of interest to readers.

However, it has a number of disadvantages:

First of all, the expert survey raises a number of questions: 

1.    Representatives of which retail fuel companies were involved in the survey? (it is necessary to explain)

Were all companies represented? (it is necessary to explain)

To what extent can a sample of 10 experts (5 experts in each target group) be considered representative?

It seems that in order to conduct an objective "pilot survey" it is necessary to involve more respondents (several representatives from each retail fuel company in each target group).

2.    It appears that the number of questions should not be limited to 4 for each category of the respondents. (There should be more questions and they should be variable)

The quality of the illustrations needs to be improved (not clear illustrations).

Author Response

Dear reviewer, 

We would like to thank you for taking the necessary time and effort to review the manuscript. We sincerely appreciate all your valuable comments and suggestions, which helped us in improving the quality of the manuscript.

We would like to reply to the recommendations as per report form:

1. We agree with the reviewer that it is necessary to specify which representatives and retail fuel companies were involved in the survey and that's why we introduced the following paragraph in chapter 3, Materials and Methods:

"The group of specialists (S) consists of employees of the top 3 companies in the oil and gas industry from Romania (Omv-Petrom group, Rompetrol and Mol), with positions in the strategic department of station network development fuel sales.

The consumer group consists of people who are from the target group of the first 3 companies of fuel sales. The consumer group is built taking into account several status indicators that the first 3 companies in the oil and gas industry target as follows: age between 30-35, with above average incomes (Romania's average net monthly salary in 2021 was RON 3540 [39]) who constantly consume services and products from gas stations."

2. We agree with the reviewer that further elaborating on this point using more data (obtained by expanding the number of interview participants and the number of questions) would be helpful. We addressed this aspect in chapter 5, Discussion.  However, we believe that at this point of time (inception phase of a larger research project) the current data we have supports our arguments. 

3. We agree with the reviewer, so we decided to slightly change the figures format. 

Once again, thank you! 

Reviewer 2 Report

Advantages:

The aim of this paper is to determine how the Romanian oil and gas industry reacts and how sensitive it is to strategic changes within organizations, with special emphasis on the technological component. The paper fills the theoretical gaps with practical implications based on interviews (although a small sample, this kind of preliminary research shows the current situation within Romanian organizations in the domain of the oil and gas industry). The paper states the importance of changing the way organizations think and act, emphasizing strategic thinking and a digital strategy that includes the application of various technologies and their impact on business security and resilience.   

At the beginning of the paper, the authors give a brief overview of the impact of globalization, the coronavirus pandemic and the current war in Ukraine on the oil and gas industry, emphasizing the situation in Romania, as the introduction. Also, they emphasize the importance of strategic thinking and the implementation of digital technologies as support for business processes. After the introductory part, the authors mention strategic changes, trends in the oil and gas industry, and strategic changes in Romanian fuel retail networks. The methodology and method of data collection are clearly explained. The results are clearly presented. After the results, a discussion and conclusion follow. The used references are appropriate and cover the research area.

Weaknesses:

The manuscript is clearly written, it is relevant to the field, however, it is necessary to work on the structure of the paper. Since the authors mention research limitations within the discussion chapter, I suggest that the discussion be renamed research limitations (and future work) and that the chapter be moved after the conclusion chapter.

I suggest:

- that the authors answer the research questions more clearly and indicate answers in the paper;

- that the authors clearly indicate the scientific contribution of the research within the paper;

- that the authors replace the current figures to make them clearer and more readable.

Author Response

Dear reviewer, 

We would like to thank you for taking the necessary time and effort to review the manuscript. We sincerely appreciate all your valuable comments and suggestions, which helped us in improving the quality of the manuscript.

We would like to reply to the recommendations as per report form:

1. We agree with the reviewer that we have to answer the research questions more clearly and indicate answers in the paper that's why we introduced the following paragraph in Conclusions

"Following the research questions stated in the paper, we can observe that the oil and gas industry is in a transition period generated by several factors mentioned in the paper, such as the Global Economic Crisis, COVID-19, the Russian invasion of Ukraine, and inflation.

Those factors challenged the industry by generating an increase in fuel pump price, a decrease in volume consumption, and changed the consumers' behaviour that forcing fuel retail companies, at the global and local level, to invest in filling station shops, new services development, digitalization for customers’ retention and attracting new ones.

In facing the challenges of these factors, oil and gas retailers must properly address them in a way as to manage to cover the new needs of the consumers.

The strategic implication of these challenges is significant, and improperly treatment can have a negative impact at the company and market level. "

2. We agree with the reviewer to replace the current figures, so we decided to change the format to make them clearer and more readable. 

Once again, thank you! 

Round 2

Reviewer 1 Report

No comments now.